# Trends and predictors of change of unmet need for family planning among reproductive age women in Ethiopia, based on Ethiopian demographic and health surveys from 2005–2016: Multivariable decomposition analysis

Abiyu Abadi Tareke[1]*, Ermias Bekele Enyew[2], Berhanu Fikadie Endehabtu[3], Abiy Tasew Dubale[2], Habitu Birhan Eshetu[4], Sisay Maru Wubante[3]

1 Deparment of Monitoring and Evaluation, West Gondar Zonal Health Department, Gondar, Ethiopia, 2 Department of Health Informatics, Mettu University, Mettu, Ethiopia, 3 Department of Health Informatics, University of Gondar, Gondar, Ethiopia, 4 Department of Health Education and Behavioral Sciences, Institute of Public Health, College of Medicine and Health Sciences, University of Gondar, Gondar, Ethiopia

* abiyu20010@gmail.com

## Abstract

### Background

By spacing births and preventing unintended pregnancies, family planning is a crucial technique strategy for controlling the fast expansion of the human population. It also improves maternal and child health. women who are thought to be sexually active but who do not use modern contraception methods, who either do not want to have any more children (Limiting) or who want to delay having children for at least two years are considered to have an unmet need for family planning (Spacing).

### Objective

This study carried out to determine which socio-demographic factors are the key contributors to the discrepancies in the unmet need for family planning among women of reproductive age between surveys years 2005 and 2016.

### Methods

The data for this study arrived from the Ethiopia Demographic Health Surveys in 2005, 2011, and 2016 to investigate trends and Predictors of change of unmet need for family planning among reproductive age women in Ethiopia. Pooled weighted sample of 26,230 (7761 in 2005, 9136 in 2011 and 9,333 in 2016 Ethiopian demographic health surveys) reproductive-age women were used for this study. For the overall trend (2005–2016) multivariable decomposition analysis for non-linear response outcome was calibrated to identify the factors that contributed to the change of unmet need for family planning. The Logit based multivariable decomposition analysis utilizes the output from the logistic regression

**Data Availability Statement:** Permission for data access was obtained from major demographic and health surveys through the online request from http://www.dhsprogram.com. The data used for this study was publicly available with no personal identifier. Our study was based on secondary data from the Ethiopian Demographic and Health survey and we have secured the permission letter from major the Demographic Health and survey. All methods were performed in accordance with the relevant guidelines and regulations of the National Ethics Review Committee of the Ethiopia Science and Technology Commission in Addis Ababa, Ethiopia and the ORC Macro Institutional Review Board in Calverton, USA. The data for this study was from 2005, 2011, 2016 Ethiopian Demographic Health Surveys (EDHS) to investigate trends and Predictors of change of unmet need for family planning among reproductive-age women in Ethiopia. The surveys were conducted using crossectional study design and employed a two-stage cluster sampling method. In the first stage, 540 Enumeration Areas (EAs) in EDHS 2005, 624 EAs for EDHS 2011, and 645 EAs in EDHS 2016 were randomly selected proportional to their EA size and, on average, 27 to 32 households per EAs were selected in the second stage. A pooled weighted sample of 26,230 (7,761 in EDHS 2005, 9,136 in EDHS 2011 and 9,333 in EDHS 2016) reproductive-age women were included for this study. The detailed information about sampling procedures was presented in the EDHS report[19, 20].

**Funding:** The author(s) received no specific funding for this work.

**Competing interests:** The authors have declared that no competing interests exist.

**Abbreviations:** EAs, enumeration areas; EDHS, Ethiopian Demographic Health Surveys; FP, family planning; SNNP, south nations and nationalities people.

model to assign the observed change in unmet need for family planning over time into two components. Stata version 16.0 was used to analyze the data.

## Result

The percentage of Ethiopian women of reproductive age who still lack access (unmet need) for family planning declined from 39.6% in 2005 to 23.6 percent in 2016. The decomposition analysis revealed that the change of unmet need for family planning was due to change in characteristics and coefficients. The difference in coefficients accounted for around nine out of 10 variations in unmet family planning need. Education level, birth order, and desired number of children were all factors that changed over the course of the last 11 years in relation to the unmet need for family planning.

## Conclusion

Between 2005 and 2016, there were remarkable changes in unmet need for family planning. Women with birth orders of five and up, women with secondary education, and women who wanted fewer children overall were the main causes of the change in unmet need for family planning.

## Introduction

By preventing undesired pregnancies and spacing births, family planning is a major strategy for slowing down population increase and enhancing mother and child health [1]. Both fecund reproductive-age women who are married and in consensual marriage have an unmet need for family planning and are believed to be sexually involved but are not using any modern contraception, either do not want to have more children (Limiting) or want to delay their next birth for at least two years, (Spacing) [2, 3]. Globally, 14% of married women had unmet family planning needs in 2015 [4].

In a developing countries like Ethiopia, approximately 225 million people had an unmet need for modern contraception. According to a systematic review study, the prevalence of unmet family planning needs in Ethiopia ranged from 26.52 to 36.39 percent, which is higher than the results of the 2016 Ethiopian Demographic and Health Survey, which revealed a prevalence of 23 percent [5]. Other studies also show high level of unmet need like study done in southwest Ethiopia (26.1%) [6], a systematic review among HIV positive women(25%) [7], Eastern Ethiopia (33%) [8] and central Ethiopia (26) [9]. Those findings suggest that, further study is required to bring down the problem. In respective researchers have looked in to a number of variables that affect unmet need for family planning, including age, parity, and religion [10], Discussions with partners and wellness extension staff, as well as awareness of contraceptive methods [11], A visit to a health center, media exposure, a husband and wife's educational status, and residence [12], Due to contraception-related factors like availability, accessibility, affordability, and side effects [13], early marriage, wealth index [14], Number of children alive, use of contraceptive methods [15], partner's attitude toward the use of family planning, current menstrual status, healthcare providers visit and discussion about family planning issues [16].

Unmet need for family planning can have serious consequences for women and their families, such as unsafe abortion, physical violence, and a high fertility rate linked to poverty and

poor maternal and child health [17, 18]. Though, improving family planning (FP) access has its own foundation for the achievement of sustainable development goal (SDG). It is also linked to human rights, gender equality and women's empowerment as well as it has negative impact on maternal, newborn, child and adolescent health. Different researchers in Ethiopia have identified the prevalence and determinant factors of unmet need to family planning. According to our deep literature reviews, studies concerning the trend and the contributing factors for the change in unmet need for FP are limited in Ethiopia. To identify factors that decrease/increase the unmet need for family planning and to aid in the development of policies and programs that are centered on lowering unmet need for family planning in Ethiopia, multivariable decomposition analysis is the most recommended and appropriate advanced statistical model to answer such kind of research questions.

## Method and materials

### Study design and sampling procedures

Using dataset of 2005, 2011, and 2016 Ethiopian Demographic Health Surveys (EDHS), this study looked at trends and predictors of change in the unmet demand for family planning among Ethiopian women of reproductive age. Those surveys were conducted using cross sectional study design and through the application of a two-stage cluster sampling method. In the first stage, 540 Enumeration Areas (EAs) in EDHS 2005, 624 EAs for EDHS 2011, and 645 EAs in EDHS 2016 were randomly selected proportional to their EA size and in the second stage, on average 27 to 32 households from each containing EA were selected. For this study, a pooled weighted sample of 26,230 (7,761 in EDHS 2005, 9,136 in EDHS 2011 and 9,333 in EDHS 2016) reproductive-age women were utilized. The detailed information about sampling procedures of the survey is presented at each EDHS report [19, 20].

### Study variables

**Outcome variable.** The outcome variable is an unmet need for FP, which is composed of an unmet need for spacing and limiting. Unmet need refers to the proportion of women who desired to either delay the current or next pregnancy or limit future pregnancies but not using any method of the modern contraception [21]. The outcome variable was categorized as "unmet need" if women had unmet need either for spacing or for limiting purpose were coded as 1, while those using FP methods for spacing or limiting or with no unmet need were "met need" coded as 0.

**Independent variables.** The independent variables included in this study were: respondent's age, respondent's educational status, religion, husband's education status, marital status, place of residence, women working status, husband working status, wealth status, media exposure, termination of pregnancy, knowledge about family planning, visited health facility last 12 months, visited by field workers in the last 12 months, perceived distance to health facility, age at first marriage, birth order, sex of household head, region and desired number of children.

**Operational definitions.** *Knowledge about family planning*. According to EDHS, having good knowledge to FP is defined as, Percentage of all respondents, currently married respondents, and sexually active unmarried respondents aged 15–49 who have heard of any contraceptive method, according to specific method.

### Statistical analysis

Important variables were extracted from the Individual Record (IR) dataset. Sample of each DHS were weighted using the "svyset" STATA command and it was applied for each

descriptive analyses. The weight variable (v005), primary sampling unit (v021), and strata (v023) are the variables required to develop the "svyset" command. Trend and decomposition analysis of the unmet need for family planning was done. The trend analysis has been done by separating based on period as (2005–20011), (2011–2016) and the overall trend (2005–2016).

For the overall trend (2005–2016) multivariable decomposition analysis for non-linear response outcome was calibrated to identify the factors that contributed to the change of unmet need for family planning across the two surveys. For our study, Logit based decomposition analysis was employed. The Logit based multivariable decomposition analysis utilizes the output from the logistic regression model to assign the observed change in unmet need for family planning over time into components.

For our study, the 2016 EDHS data was appended to the 2005 EDHS data using the "append" Stata command, and the Logit based multivariable decomposition analysis (using mvdcmp STATA command) was used to identify factors that contributed to the change in unmet need for family planning over the last 11 years. The change in unmet need for family planning can be explained by the compositional difference between surveys (i.e. differences in characteristics) and/or the difference in effects of explanatory variables (i.e. differences in the coefficients) between the surveys. Hence, the observed decrease in unmet need over time is additively decomposed into a compositional difference of respondents of each survey (endowments) component and a coefficient (or effects of characteristics) component.

For logistic regression, the Logit or log-odd of unmet need for family planning is taken as:

$$\text{Logit}(2005) - \text{Logit}(2016) = F(X2005\beta2005) - (F\, X2016\beta2016)$$

$$= \frac{[\mathbf{F(X2005\ \beta2005) - F(X2016\ \beta2005)}]}{\mathbf{E}} + \frac{[\mathbf{F(X2016\ \beta2005) - F(X2016\ \beta2016)}]}{C}$$

[22]

X indicates independent variables (unmet need for FP in this study)

β denotes that, the regression coefficient of each selected contributing variables

The E component refers to the part of the differential owing to differences in endowments or characteristics. The C component refers to that part of the differential attributable to differences in coefficients or effects.

## Ethical approval and consent

Authors have requested DHS Program through an online request by written letter of objective and significance of the study. Permission for data access was granted to download and use the data from http://www.dhsprogram.com. The EDHS programs permitted data access, and data were used for only the current study.

## Result

### Characteristics of the study population

Table 1 below illustrates the percentage distribution of selected characteristics of respondents in the 2005, 2011 and 2016 Ethiopian Demographic and Health Surveys. It is apparent that women aged 25–34 years were the dominant percentage of women across the three successive surveys. Across the three surveys, there was a clear trend of decreasing the percentage of unmet need for spacing (from 25.5% to 14.34%) and limiting (from 16.05% to 9.22%) by 6.83% point values. Regarding the educational status of the study participants, women with no education decreased by 17.2% in 11 years. However, the number of women with primary school and those with high school and above increased by 12.7 and 4.6 percent, respectively.

**Table 1. Percentage distribution of socio-demographic characteristics among respondents, 2005, 2011 and 2016 EDHS.**

| Characteristics | 2005 EDHS N = 7,761 | 2011 EDHS N = 9,136 | 2016 EDHS N = 9,333 | 2011–2005 | 2016–2011 | 2016–2005 |
|---|---|---|---|---|---|---|
| **Unmet need of family planning** | | | | | | |
| Unmet need for spacing | 23.5% | 18.81% | 14.34% | -4.69% | -4.47% | -9.16% |
| Unmet need for limiting | 16.05% | 10.18% | 9.22% | -5.87% | -0.96% | -6.83% |
| Total unmet need | 39.55% | 29% | 23.6% | -10.6% | -5.4% | -16% |
| **Age of respondents** | | | | | | |
| 15–24 | 28.9% | 28.2% | 26.0% | -0.7% | -2.2% | -2.9% |
| 25–34 | 43.9% | 45.2% | 46.6% | 1.3% | 1.4% | 2.7% |
| 35+ | 27.2% | 26.6% | 27.4% | -0.6% | 0.8% | 0.2% |
| **Region** | | | | | | |
| Tigray | 6.5% | 6.5% | 6.8% | 0.0% | 0.3% | 0.3% |
| Afar | 1.1% | 1.0% | 0.9% | -0.1% | -0.1% | -0.2% |
| Amhara | 25.2% | 26.3% | 24.7% | 1.1% | -1.6% | -0.5% |
| Oromia | 36.9% | 38.2% | 38.3% | 1.3% | 0.1% | 1.4% |
| Somali | 3.8% | 2.2% | 2.9% | -1.6% | 0.7% | -0.9% |
| Benishangul-gumuz | 1.0% | 1.2% | 1.1% | 0.2% | -0.1% | 0.1% |
| SNNP | 22.0% | 19.7% | 20.5% | -2.3% | 0.8% | -1.5% |
| Gambela | 0.3% | 0.5% | 0.3% | 0.2% | -0.2% | 0.0% |
| Harari | 0.3% | 0.3% | 0.2% | 0.0% | -0.1% | -0.1% |
| Addis Ababa | 2.7% | 3.9% | 3.9% | 1.2% | 0.0% | 1.2% |
| Dire-Dawa | 0.4% | 0.4% | 0.5% | 0.0% | 0.1% | 0.1% |
| **Partner's educational status** | | | | | | |
| No Education | 56.7% | 46.4% | 44.4% | -10.3% | -2.0% | -12.3% |
| Primary | 29.5% | 41.0% | 38.1% | 11.5% | -2.9% | 8.6% |
| Secondary& above | 13.2% | 11.8% | 16.9% | -1.4% | 5.1% | 3.7% |
| Orthodox | 45.9% | 44.5% | 42.1% | -1.4% | -2.4% | -3.8% |
| **Religion** | | | | | | |
| Catholic | 1.2% | 1.0% | 0.7% | -0.2% | -0.3% | -0.4% |
| Protestant | 18.9% | 22.3% | 21.6% | 3.4% | -0.7% | 2.7% |
| Muslim | 31.7% | 30.4% | 33.9% | -1.3% | 3.5% | 2.2% |
| Traditional | 1.4% | 0.9% | 1.0% | -0.5% | 0.2% | -0.4% |
| Other | 1.0% | 0.9% | 0.7% | -0.1% | -0.2% | -0.3% |
| **Respondent's educational status** | | | | | | |
| No Education | 75.8% | 61.8% | 58.6% | -14.0% | -3.2% | -17.2% |
| Primary | 17.0% | 30.4% | 29.7% | 13.4% | -0.7% | 12.7% |
| Secondary& above | 7.2% | 7.8% | 11.8% | 0.6% | 4.0% | 4.6% |
| **Wealth status** | | | | | | |
| Poorest | 19.3% | 20.0% | 19.1% | 0.7% | -0.9% | -0.2% |
| Poorer | 20.6% | 20.1% | 20.2% | -0.5% | 0.1% | -0.4% |
| Middle | 21.3% | 19.6% | 20.1% | -1.7% | 0.5% | -1.2% |
| Richer | 19.7% | 18.7% | 19.1% | -1.0% | 0.4% | -0.6% |
| Richest | 19.1% | 21.6% | 21.6% | 2.5% | 0.0% | 2.5% |
| **Place of residency** | | | | | | |
| Urban | 11.1% | 19.1% | 17.0% | 8.0% | -2.1% | 5.9% |
| Rural | 89.0% | 80.9% | 83.0% | -8.1% | 2.1% | -6.0% |
| **Sex of household head** | | | | | | |
| Male | 90.7% | 86.3% | 85.5% | -4.4% | -0.8% | -5.2% |

*(Continued)*

**Table 1.** (Continued)

| Characteristics | 2005 EDHS N = 7,761 | 2011 EDHS N = 9,136 | 2016 EDHS N = 9,333 | 2011–2005 | 2016–2011 | 2016–2005 |
|---|---|---|---|---|---|---|
| Female | 9.7% | 12.7% | 14.5% | 3.0% | 1.8% | 4.8% |
| **Birth order** | | | | | | |
| 1st | 14.76% | 16.04% | 16.67% | 1.3% | 0.6% | 1.9% |
| 2nd | 14.79% | 17.15% | 15.50% | 2.4% | -1.7% | 0.7% |
| 3rd | 14.11% | 14.21% | 15.06% | 0.1% | 0.8% | 1.0% |
| 4th | 13.15% | 12.82% | 12.91% | -0.3% | 0.1% | -0.2% |
| 5th & above | 43.19% | 39.78% | 39.87% | -3.4% | 0.1% | -3.3% |
| **Age at first marriage** | | | | | | |
| <18 years | 69.8% | 64.4% | 62% | -5.4% | -2.4% | -7.8% |
| > = 18 years | 30.2% | 35.6% | 38% | 5.4% | 2.4% | 7.8% |
| **Working status** | | | | | | |
| Not working | 68.8% | 43.7% | 49.25% | -25.1% | 5.6% | -19.6% |
| Working | 31.2% | 56.3% | 50.75% | 25.1% | -5.6% | 19.6% |

Percentage of orthodox Christian declined by 1.4% and 2.4% from 2005 to 2011 and 2011 to 2016 respectively. But, the percentage of Protestants and Muslim followers increased by 2.7% and 2.2% from 2005 to 2016 surveys respectively.

Concerning households' wealth status, scanty change was occurred between the period of 2005 and 2016. Poorest, poorer, middle and richer shows little reduction ranging from 0.2% to 1.2% point values. But, households with the richest category shows relatively highest rise i.e. 2.5 point percentage over the three study periods.

## Trends of unmet need for FP

In the last three successive demographic health surveys, the unmet need for family planning declined from 39.6% in 2005 to 23.6% in 2016 i.e. 16 percent. The second highest rate of decline was observed from 2005 (39.6%) to 2011 (29%) i.e. about 10.6 percent of change next to 2005 to 2016 time period. 5.4% point of fall was also noticed from 2011(29%) to 2016 (23.6%) (Fig 1). Overall a significant change (not overlapping 95% confidence interval) was observed across the three period of study i.e. 2005 to 2011, 2011 to 2016 and 2005 to 2016 EDHS.

Regionally, Oromia showed the largest decline in the proportion of unmet need for family planning i.e. 5.6% fall. Next to Oromia region, SNNP (by 5%), followed by Amhara region (4.7%) showed reduction in unmet need for FP. Even though, between 2005 and 2016 the amount of unmet need diminished by 16 percent, there has been a steady decrement of unmet need in urban areas i.e.by 0.1% point percentage. Regarding religion relatively highest reduction in unmet need for family planning was shown among orthodox, protestant and Muslim followers from 2005 to 2016 by 9%, 4% and 2.2% respectively (Table 2).

Additionally, there was decrease in the composition of unmet need for family planning in women who had no education from 2005 to 2016 at 16.2% point percentage drop. On the other hand, from 2005 to 2016, there was a marginal increase, or a 0.3 percent increase, in the number of women who completed high school educational programs. For 11 years, women with only a primary education had a nearly continuous trend of unmet family planning needs. As birth order (parity) increase the proportion of unmet need for family planning also increases. In 2005 EDHS huge difference in unmet need was noticed between women having

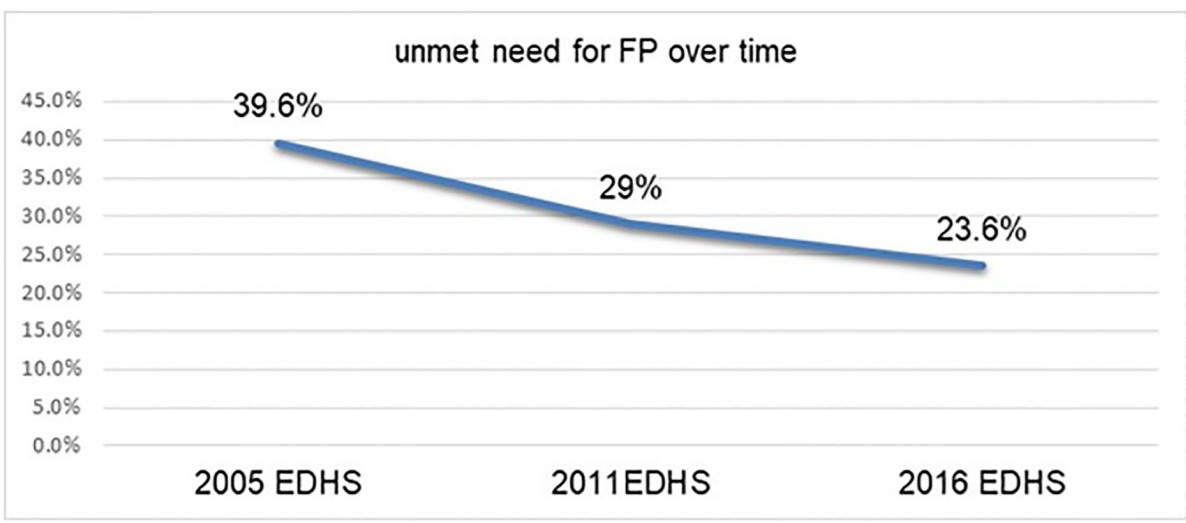

**Fig 1. The trend in the rate of unmet need for family planning among reproductive-age women in Ethiopia from 2005 to 2016 surveys.**

1st birth order and having 5 and above birth, which was 4.8% to 20.4% respectively. Similar pattern of difference was also occurred in 2016 EDHS i.e. 2.4% in first birth order and 13.8% in women having five and above birth history. Also, there have been a similar drop in the prevalence of unmet need over the last one decade years in every category of wealth quintile groups.

## Overall decomposition analysis

Variables with p-value less than 0.2 from the bivariable decomposition analysis were nominated as candidate variables for multivariable decomposition analysis. Compositional factors including region, religion and place of residency were excluded from multivariable compositional analysis list because having p-value greater than 0.2.

 **Difference due to characteristics (Endowment).** Generally, there has been a decrement in the unmet need for family planning among reproductive age group women in Ethiopia from 2005 to 2016. The multivariable decomposition analysis result showed that about 9% of the change in unmet need for family planning among reproductive age group women was explained by differences in respondent's characteristics (endowment) between the two surveys (Table 3). Among the various important compositional factors, such as wealth index, birth order (parity), place of residence and perceived distance from health facility had a significant contribution to the change of unmet need for family planning.

 Perceived distance from health facility was the individual characteristic that had the biggest impact on the change in unmet need for FP between 2005 and 2016. This indicates that a 6.4 percent increase in the unmet need for FP is due to a drop in the number of women who do not consider travel time to a health institution to be a major issue. The second largest characteristic effect on the observed change in unmet need was due to the decrement of women who have higher number of children (five and above) i.e. explains about 5% increase in unmet need. Similarly, a 2.5% increment in unmet need for family planning was due to the decrement in the composition of women who reside in rural areas (Table 4).

 **Difference due to effects of the coefficient.** Holding the effect of change in compositional characteristics constant, about nine in ten changes in unmet need for family planning was attributable to the difference in coefficients (Table 3). This means the change in unmet

**Table 2. Trends of unmet need for family planning among reproductive age from 2005, 2011 and 2016 EDHS.**

| Characteristics | 2005 EDHS N = 7,761 | 2011 EDHS N = 9,136 | 2016 EDHS N = 9,333 | 2011–2005 | 2016–2011 | 2016–2005 |
|---|---|---|---|---|---|---|
| **Age of respondents** | | | | | | |
| 15–24 | 10.5% | 7.2% | 4.8% | -3.3% | -2.4% | -5.7% |
| 25–34 | 16.8% | 12.3% | 10.3% | -4.5% | -2.0% | -6.5% |
| 35+ | 12.3% | 9.5% | 8.4% | -2.9% | -1.1% | -3.9% |
| **Region** | | | | | | |
| Tigray | 1.77% | 1.57% | 1.21% | -0.20% | -0.36% | -0.56% |
| Afar | 0.19% | 0.18% | 0.17% | -0.01% | -0.01% | -0.02% |
| Amhara | 8.96% | 6.81% | 4.25% | -2.15% | -2.56% | -4.71% |
| Oromia | 17.63% | 13.14% | 12.06% | -4.49% | -1.08% | -5.57% |
| Somali | 0.55% | 0.61% | 0.37% | 0.06% | -0.24% | -0.18% |
| Benishangul-Gumuz | 0.36% | 0.34% | 0.24% | -0.02% | -0.10% | -0.12% |
| SNNP | 9.60% | 5.61% | 4.60% | -3.99% | -1.01% | -5.00% |
| Gambella | 0.10% | 0.09% | 0.07% | -0.01% | -0.02% | -0.03% |
| Harari | 0.06% | 0.08% | 0.05% | 0.01% | -0.02% | -0.01% |
| Addis Ababa | 0.32% | 0.47% | 0.43% | 0.15% | -0.04% | 0.11% |
| Dire-Dawa | 0.07% | 0.10% | 0.11% | 0.02% | 0.02% | 0.04% |
| **Partner's educational status** | | | | | | |
| No Education | 22.9% | 14.8% | 12.1% | -8.14% | -2.64% | 10.8% |
| Primary | 13.0% | 12.4% | 9.1% | -0.64% | -3.25% | -3.9% |
| Secondary& above | 3.9% | 2.0% | 2.8% | -1.9% | 0.8% | -1.2% |
| **Religion** | | | | | | |
| Orthodox | 16.8% | 11.1% | 7.7% | -5.7% | -3.4% | -9.0% |
| Catholic | 0.6% | 0.4% | 0.3% | -0.3% | -0.1% | -0.4% |
| Protestant | 8.3% | 6.5% | 4.3% | -1.8% | -2.1% | -3.9% |
| Muslim | 12.8% | 10.6% | 10.6% | -2.3% | 0.1% | -2.2% |
| Traditional | 0.6% | 0.3% | 0.4% | -0.3% | 0.1% | -0.2% |
| **Respondent's educational status** | | | | | | |
| No Education | 31.6% | 19.5% | 15.4% | -12.1% | -4.1% | -16.2% |
| Primary | 6.7% | 8.7% | 6.6% | 2.0% | -2.0% | 0.0% |
| Secondary& above | 1.3% | 0.9% | 1.5% | -0.4% | 0.7% | 0.3% |
| **Wealth status** | | | | | | |
| Poorest | 7.5% | 7.0% | 5.3% | -0.5% | -1.7% | -2.2% |
| Poorer | 9.2% | 6.2% | 5.7% | -3.0% | -0.5% | -3.5% |
| Middle | 9.1% | 6.5% | 5.0% | -2.6% | -1.5% | -4.1% |
| Richer | 8.5% | 5.8% | 4.3% | -2.7% | -1.5% | -4.2% |
| Richest | 5.2% | 3.6% | 3.2% | -1.7% | -0.4% | -2.0% |
| **Place of residency** | | | | | | |
| Urban | 2.1% | 3.2% | 2.1% | 1.0% | -1.1% | -0.1% |
| Rural | 37.4% | 25.9% | 21.5% | -11.5% | -4.4% | -16.0% |
| **Sex of household head** | | | | | | |
| Male | 36.0% | 25.2% | 20.0% | -10.8% | -5.1% | -16.0% |
| Female | 3.6% | 3.8% | 3.5% | 0.3% | -0.3% | 0.0% |
| **Birth order** | | | | | | |
| 1st | 4.8% | 3.3% | 2.4% | -1.5% | -0.9% | -2.4% |
| 2nd | 4.5% | 3.8% | 2.6% | -0.7% | -1.2% | -2.0% |
| 3rd | 5.7% | 3.5% | 2.9% | -2.2% | -0.6% | -2.8% |

*(Continued)*

**Table 2.** (Continued)

| Characteristics | 2005 EDHS N = 7,761 | 2011 EDHS N = 9,136 | 2016 EDHS N = 9,333 | 2011–2005 | 2016–2011 | 2016–2005 |
|---|---|---|---|---|---|---|
| 4th | 5.0% | 3.5% | 2.9% | -1.4% | -0.7% | -2.1% |
| 5th & above | 20.4% | 15.6% | 13.8% | -4.8% | -1.7% | -6.6% |
| **Age at first marriage** | | | | | | |
| <18 years | 11.8% | 8.9% | 7.9% | -2.9% | -1.0% | -3.9% |
| > = 18 years | 28.1% | 20.2% | 15.7% | -7.8% | -4.5% | -12.3% |
| **Working status** | | | | | | |
| Not working | 12.0% | 15.8% | 11.3% | 3.8% | -4.5% | -0.7% |
| Working | 27.6% | 13.2% | 12.3% | -14.4% | -0.9% | -15.3% |

need for family planning among reproductive-age women was explained by differences in coefficient (effects of characteristics) across the two surveys. Factors associated with the change of unmet need for family planning over the last 11 years were educational status, birth order, and desired number of children. Among these, high birth order made the largest contribution to the decrement in unmet need for family planning, accounting for more than one-fourth of the changes [Coefficient = 0.04756, P< 0.01].

However, being a woman with less than five ideal number of desired children was the largest positive (increase) contributor through uplifting unmet need for family planning nearly by fifth point percentage (20%). Another interesting finding was that Women who had secondary education showed a significant negative contribution to the observed percentage decrease in unmet need for family planning over the study period which contributed about 5.5% with [Coefficient = 0.0092, P< 0.05]. In other word being educated are increasingly displaying substantial influence in the reduction of unmet need for family planning (Table 4).

## Discussion

This study demonstrated factors which contributed to the change of unmet need for FP from 2005 to 2016 surveys in Ethiopia. The trend of unmet need for FP was decreased by 16% from 2005 to 2016 surveys. Majority of the change was contributed by change of coefficients (i.e.91%).

Respondent's educational level, wealth status, place of residency, parity and perceived distance from health facility were the contributory factors for the overall change of unmet need for family planning between 2005 and 2016. This finding Are in agreement with previous studies done in Ethiopia and Pakistan [23, 24]. The expansion of health extension programs in Ethiopia in the last 15 years might have had its own contribution to the decrement of unmet need for FP. There is also documented evidence of improvement in women's autonomy in making decisions for their own health [25, 26].

**Table 3. Summary of overall decomposition results of unmet need for family planning in Ethiopia 2005 to 2016 EDHS.**

| Unmet need for family planning | Coefficient | p-value | 95% CI | Percent |
|---|---|---|---|---|
| E | -0.0147 | 0.000 | (-0.022, -0.008) | 8.93 |
| C | -0.1504 | 0.000 | (-0.174, -0.126) | 91.06 |
| R | -0.1652 | 0.000 | (-0.188, -0.143) | |

E = difference as a result of Endowment; C = difference as a result of coefficients; R = Residual; High outcome group: year = = 1- - -Low outcome group: year = = 0.

**Table 4. Decomposition of change in unmet need for family planning in Ethiopia, 2005 to 2016.**

| Characteristics | Characteristics effect (E) | | f coefficients effect (C) | |
|---|---|---|---|---|
| | Coefficient | Share (%) | Coefficient | Share (%) |
| **Age of respondents** | | | | |
| 15–24 | 0 | | 0 | |
| 25–34 | -0.00100 | 0.60637 | 0.01610 | -9.75 |
| 35+ | -0.00057 | 0.34577 | 0.01105 | -6.69 |
| **Respondent's educational status** | | | | |
| None | 0 | | 0 | |
| Primary | 0.00457 | -2.7682 | 0.00978 | -5.91 |
| Secondary | 0.000139 | -0.084 | **0.0091**$^*$ | -5.50 |
| Higher | 0.00186 | -1.127 | .0018788 | -1.14 |
| **Wealth status** | | | | |
| Poor | 0 | | 0 | |
| Middle | 0.00033 | -0.19948 | -0.0063 | 3.83 |
| Rich | **-0.00069**$^*$ | 0.41661 | -0.01811 | 11.38 |
| **Age at marriage** | | | | |
| > = 18 years | 0 | | 0 | |
| <18 years | .001422 | -.86062 | .0013496 | -.81681 |
| **Place of residency** | | | | |
| Urban | 0 | | 0 | |
| Rural | **-0.00010**$^*$ | 2.4612 | -.039155 | 23.70 |
| **Birth order** | | | | |
| 1$^{st}$ | 0 | | 0 | |
| 2$^{nd}$ | 0.00008 | -0.051711 | 0.00647 | -3.92 |
| 3$^{rd}$ | 0.00070 | -0.42806 | -0.00056 | 0.34 |
| 4$^{th}$ | **-0.00074**$^*$ | 0.451 | 0.01124 | -6.80 |
| 5$^{th}$ & above | **-0.0084**$^{***}$ | 5.101 | **0.045756**$^{**}$ | -27.69 |
| **Desired no. of children** | | | | |
| 5+ | 0 | | 0 | |
| <5 | -0.00024 | 0.14378 | **-0.03290**$^{**}$ | 19.91 |
| **Perceived distance from health facility** | | | | |
| big problem | | | 0 | |
| Not big problem | **-0.01059**$^{***}$ | 6.4108 | -0.01289 | 7.80 |
| Constant | - - - | - - - - - - | -0.1492 | 90.30 |

$^*$: p-value <0.05,

$^{**}$: p-value<0.01 &

$^{***}$: p-value <0.001.

According to the study's findings, rural inhabitants have had a greater drop in unmet family planning needs than urban ones (Table 2). This may be a result of the government's efforts over the past ten years to increase rural communities' awareness of maternal and child health issues and the availability of healthcare services. According to the decomposition analysis, the proportion of women who had five or more children, thought that traveling a long way to a health facility wasn't a big deal, and who lived in rural areas and came from wealthy households contributed to the rise in the proportion of women with unmet family planning needs. Over the course of the whole study period, the proportion of women who believe that traveling to a health facility is not a major issue has decreased significantly, which has a considerable

impact on the rise in unmet need for FP (2005–2016). This may be attributed to Ethiopia's improvement in health facilities' physical accessibility and affordability over recent decades. In addition, specially trained community health workers (health extension workers) and the availability of health posts to the nearby community had their own contribution to the enhancement of maternal health services such as family planning [27].

Similarly, women with para five and above have a significantly higher unmet need for FP when compared to women with para one. Women with a high number of children are more likely to face the unmet need for FP because of having too many children, unlikely to fear child death, and because they think of themselves as reaching the planned level of fertility.

From the 2005 to 2016 surveys, the compositional decrement of rural residence raised the unmet need for FP by 2.5% point percentage relative to an urban resident. This can be explained by the speedy urbanization over the last decades [28]. Urbanization was important in enhancing access to health facilities and having a higher knowledge of maternal health service use than rural residents.

Change due to coefficient differences between the two surveys having secondary education, having many children (the highest birth order) and the ideal number of desired children were significantly associated with the change of unmet need for FP. About a 28% decrease in unmet need for FP was attributable to women having 5 or above birth orders. This finding is consistent with the study done in Ethiopia [29].

Women who wanted fewer than five children contributed 20% more to the increase in unmet need for family planning than women who wanted more than five children, a finding that is consistent with another recent study in Ethiopia [28]. This can be explained by; women who desired to have less than five children being more likely to face challenges related to unmet need for FP to limit their number of children below the desired number.

Another finding was that having secondary education contributed to decreasing the unmet need by 5.5%, similar to what has been documented in other studies in Ethiopia and Kenya [30, 31]. Women with a secondary education may have more access to knowledge about family planning, or formal education may have allowed them to have a better understanding of contraception [32]. These educated women were also more likely to make their own family planning decisions [33]. The findings of the study may inform maternal health programmers to strengthen home visits by health care workers to improve family planning uptake.

These findings could help policymakers identify and prioritize interventional strategies based on the core contributing factors. For example, if educational status has a major share of the change of unmet need compared to other factors, policy maker will be fruitful if they design interventional plans focusing on education rather than wasting resources on other strategies. As this research utilized the new model in the era of health, it helps researchers to replicate it. Even though the authors compared three large data sets to show the trend and contributing factors to the change of unmet need, they did not consider other significant contributing variables (cultural, clinical, and other factors) which were not collected by the EDHS program. Additionally, this research is not free from recall bias because during the survey time period, women were asked to about condition of their socio-demographic characteristics over the last 5 years prior to survey.

## Conclusion

A remarkable change in unmet need for FP was observed between the periods 2005 and 2016. Both changes in characteristics and coefficient were the contributing elements to the observed change in unmet need for FP. The majority of the change in unmet need for FP was due to differences in coefficient over the study period. Mainly, the change in unmet need for FP was due

to a change in women's having birth orders of five or above, having secondary education, and women who desired the number of children below five. Empowering uneducated women about maternal health services, specifically about family planning, is required. The government and any concerned body could better focus on the enhancement of household economic status and health facility accessibility.

## Author Contributions

**Conceptualization:** Abiyu Abadi Tareke, Ermias Bekele Enyew.

**Data curation:** Abiyu Abadi Tareke, Ermias Bekele Enyew.

**Formal analysis:** Abiyu Abadi Tareke, Ermias Bekele Enyew.

**Funding acquisition:** Abiyu Abadi Tareke.

**Investigation:** Abiyu Abadi Tareke, Ermias Bekele Enyew, Berhanu Fikadie Endehabtu.

**Methodology:** Abiyu Abadi Tareke, Ermias Bekele Enyew.

**Project administration:** Abiyu Abadi Tareke.

**Resources:** Abiyu Abadi Tareke, Ermias Bekele Enyew, Berhanu Fikadie Endehabtu.

**Software:** Abiyu Abadi Tareke, Ermias Bekele Enyew, Habitu Birhan Eshetu.

**Supervision:** Abiyu Abadi Tareke.

**Validation:** Ermias Bekele Enyew.

**Visualization:** Abiyu Abadi Tareke, Ermias Bekele Enyew.

**Writing – original draft:** Abiyu Abadi Tareke, Ermias Bekele Enyew, Berhanu Fikadie Endehabtu, Abiy Tasew Dubale, Habitu Birhan Eshetu, Sisay Maru Wubante.

**Writing – review & editing:** Abiyu Abadi Tareke, Ermias Bekele Enyew, Berhanu Fikadie Endehabtu, Abiy Tasew Dubale, Habitu Birhan Eshetu, Sisay Maru Wubante.

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
