## [Decision Letter · Decision Letter 0]

25 Feb 2022

PONE-D-21-17862

Trends and Predictors of Change of Unmet Need for Family Planning among Reproductive Age Women in Ethiopia , based on Ethiopian Demographic and Health Surveys from 2005-2016: Multivariate Decomposition Analysis

PLOS ONE

Dear Dr. Tareke,

Thank you for submitting your manuscript to PLOS ONE. After careful consideration, we feel that it has merit but does not fully meet PLOS ONE’s publication criteria as it currently stands. Therefore, we invite you to submit a revised version of the manuscript that addresses the points raised during the review process.

We look forward to receiving your revised manuscript.

Kind regards,

Jianhong Zhou

Associate Editor

PLOS ONE

Journal Requirements:

Reviewers' comments:

Reviewer's Responses to Questions

**Comments to the Author**

1. Is the manuscript technically sound, and do the data support the conclusions?

Reviewer #1: Yes

Reviewer #2: Yes

2. Has the statistical analysis been performed appropriately and rigorously? 

Reviewer #1: Yes

Reviewer #2: Yes

3. Have the authors made all data underlying the findings in their manuscript fully available?

Reviewer #1: Yes

Reviewer #2: Yes

4. Is the manuscript presented in an intelligible fashion and written in standard English?

Reviewer #1: Yes

Reviewer #2: Yes

5. Review Comments to the Author

Reviewer #1: This was an interesting paper, addressing an important topic using generally appropriate methods and a sound data set. However, there were a number of issues with the paper, including formulation of the models, discussion of the results, and quality of the prose and editing which mean that it would need a considerable amount of work for it to be suitable for publication.

1. One of my main areas of concern revolves around the definition of the dependent variable, lines 97-102;

“The outcome variable was unmet need for FP, where it composed of unmet need for spacing and limiting. It refers to the proportion of women who desire to either delay the next pregnancy or limit future pregnancies but are not using any method of modern method of contraception (11). The outcome variable was categorized as “unmet need” if women had unmet need for spacing and limiting were and coded as 1, while those using FP methods for spacing or limiting or with no unmet need were “met need” coded as 0”.

But to my understanding, unmet need considers not only sexually active women who desire to delay/limit future pregnancies but it includes those who were currently pregnant and gave birth recently, whether the pregnancy or recent birth was wanted or not (Look the framework below). How do you justify this?

2. My next concern is, Abebaw et al, 2015 tried to address the issue with a similar procedure and dataset see, https://journals.plos.org/plosone/article/comments?id=10.1371/journal.pone.0116525

What is your contribution?

3. The discussion would benefit from a clearer distinction between contributions to the unmet need change as found here to result from changes in 1) the characteristics of women and 2) the correlates (or effects) of these characteristics with (on) contraceptive use.

Finally, in my opinion, the manuscript could be considered for publication but prior to publication minor revisions are suggested to improve the manuscript.

Reviewer #2: Dear authors,

Unmet need for family planning is an important study to address the trends and predictors factors. This is an interesting study and the authors have conducted. The paper is generally well written and structured. However, in my opinion the paper has some issues that need to consideration. Below I have provided numerous recommendations.

Introduction:

Please focus could be on the entire Ethiopia context. It be more specific.

Methods,

The authors should provide clear the study design.

Could you please provide the clear inclusion and exclusion criteria that you developed prior to recruitment of the study participants?

Please provide the wealth status measurement.

Please provide the knowledge about family planning measurement.

Please provide the perceived distance to health facility measurement.

Could you please provide the used appropriate statistical analysis to control and adjust of the effects of confounding factors?

Discussion

Please clarify the methodological limitations of study with scouse on risk of biases.

Could you please provide application of study findings in education, policy making and future research.

6. PLOS authors have the option to publish the peer review history of their article (what does this mean?). If published, this will include your full peer review and any attached files.

Reviewer #1: No

Reviewer #2: **Yes: **Dr Zeinab Hamzehgardeshi, Associate Professor in Reproductive Health, Sexual and Reproductive Health Research Center, Mazandaran University of Medical Sciences, Sari, Iran

---

## [Author Response · Author response to Decision Letter 0]

30 Mar 2022

Author’s response to reviews

Title: Trends and Predictors of Change of Unmet Need for Family Planning among Reproductive Age Women in Ethiopia, based on Ethiopian Demographic and Health Surveys from 2005-2016: Multivariate Decomposition Analysis

Authors:

Abiyu Abadi Tareke (abiyu20010@gmail.com)

 Ermias Bekele Enyew (Ermiashi@gmail.com)

Version: 1 

Date: March 30, 2022

Author’s response to reviews: 

Plos one 

Point by point response for editors/reviewers comments

Manuscript title: Trends and Predictors of Change of Unmet Need for Family Planning among Reproductive Age Women in Ethiopia, based on Ethiopian Demographic and Health Surveys from 2005-2016: Multivariate Decomposition Analysis

Manuscript ID: PONE-D-21-17862R1

Dear editor/reviewer:

Dear all,

We would like to thank you for your substantial, enlightening and positive comments that you give us. Your valuable comments would progress the substance and content of this manuscript. We tried to account each comments and clarification questions of editors and reviewers in a focused way. Our point-by-point responses to each comment and questions are detailed on the following pages. Further, the details of changes were shown by track changes in the supplementary document attached.

Response to Editors and reviewers’ comments

Reviewer 1 comment: This was an interesting paper, addressing an important topic using generally appropriate methods and a sound data set. However, there were a number of issues with the paper, including formulation of the models, discussion of the results, and quality of the prose and editing which mean that it would need a considerable amount of work for it to be suitable for publication.

1. One of my main areas of concern revolves around the definition of the dependent variable, lines 97-102;

“The outcome variable was unmet need for FP, where it composed of unmet need for spacing and limiting. It refers to the proportion of women who desire to either delay the next pregnancy or limit future pregnancies but are not using any method of modern method of contraception (11). The outcome variable was categorized as “unmet need” if women had unmet need for spacing and limiting were and coded as 1, while those using FP methods for spacing or limiting or with no unmet need were “met need” coded as 0”.

But to my understanding, unmet need considers not only sexually active women who desire to delay/limit future pregnancies but it includes those who were currently pregnant and gave birth recently, whether the pregnancy or recent birth was wanted or not (Look the framework below). How do you justify this?

Authors’ response: thank you reviewer for rising very decisive issue here. Originally, we included women who had unintended pregnancy during the time of data collection (i.e. those with either unwanted or mistimed current pregnancy). But, we didn’t operationalized it well. We re-operationalize under the revised version of this study as “The outcome variable was an unmet need for FP, where it was composed of an unmet need for spacing and limiting. It refers to the proportion of women who desired to either delay the current or next pregnancy or limit future pregnancies but are not using any method of the modern method of contraception. The outcome variable was categorized as “unmet need” if women had unmet need for spacing and limiting were coded as 1, while those using FP methods for spacing or limiting or with no unmet need were “met need” coded as 0.” 

We have done the whole analysis based on The Ethiopian Demographic Health Surveys (EDHS) guideline. Which defined unmet need for family planning as “Proportion of women who are not pregnant and not postpartum amenorrhoeic and are considered fecund and want to postpone their next birth for 2 or more years or stop childbearing altogether but are not using a contraceptive method, or have a mistimed or unwanted current pregnancy, or are postpartum amenorrhoeic and their last birth in the last 2 years was mistimed or unwanted.”

Reviewer 1 comment: 

My next concern is, Abebaw et al, 2015 tried to address the issue with a similar procedure and dataset see, https://journals.plos.org/plosone/article/comments?id=10.1371/journal.pone.0116525

What is your contribution?

Authors’ response: thank you for your comment. Those authors didn’t incorporated the most recent EDHS datasets i.e. 2016. Additionally, they addressed the issue of modern contraception utilization which is opposite to our study concept. We want to show trend of unmet need for family planning from 2005 to 2016 and the principal contributing factors for the decrease or increase of unmet need from 2005 to 2016. Therefore, our study is different in both issue to be addressed and years of surveys of the datasets. 

Reviewer 1 comment: The discussion would benefit from a clearer distinction between contributions to the unmet need change as found here to result from changes in 1) the characteristics of women and 2) the correlates (or effects) of these characteristics with (on) contraceptive use.

Authors’ response: thank you reviewer we made clear demarcation between those two contributors accordingly. 

Reviewer #2: Dear authors,

Unmet need for family planning is an important study to address the trends and predictors factors. This is an interesting study and the authors have conducted. The paper is generally well written and structured. However, in my opinion the paper has some issues that need to consideration. Below I have provided numerous recommendations.

Introduction:

Reviewer #2 comment: Please focus could be on the entire Ethiopia context. It be more specific.

Authors’ response: thank you reviewer to your comment. We tried to concise the introduction part of this study (see the revised version of this manuscript).

Method

Reviewer #2 comment: The authors should provide clear the study design.

Authors’ response: thank you for your constructive comment. We further analyzed secondary data from Ethiopian demographic health surveys, which was conducted using crossectional method of study design. We added the study design to the revised version of this manuscript accordingly.

Reviewer #2 comment: Could you please provide the clear inclusion and exclusion criteria that you developed prior to recruitment of the study participants?

Authors’ response: thank you for your concern. As we further analyzed the 2005, 2011, and 2016 EDHS, we didn’t have the option to include or exclude study participants. This is because of the process of including and excluding study participants was held during the time of data collections. But, each EDHS reported that women who were postpartum amenorrhoeic, who were considered fecund, who were using family planning, who want to be pregnant sooner were excluded from their initial study. 

Reviewer #2 comment: Please provide the wealth status measurement.

Authors’ response: thank you for your positive comment. Wealth status related information was taken directly as it appears in the EDHS reports. But, the EDHS program calculate wealth index of each household according to their wealth in cash (number) and in kinds of consumer goods households own and housing characteristics. These scores are derived using principal component analysis (PCA). Therefore, we didn’t take any participation in measurement of this variable rather than using as it is. 

Reviewer #2 comment: Please provide the knowledge about family planning measurement.

Authors’ response: thanks for your constructive comment. This variable is operationalized and corrected accordingly (see the revised version of this manuscript). 

Reviewer #2 comment: Please provide the perceived distance to health facility measurement.

Authors’ response: thank you for your productive comment. The EDHS program didn’t measure the exact distance from interviewee’s home to a health facility. Rather, the data collector asks the study participants “what is seems perception of participants the distance to health facility”. In particular, Women were asked whether the distance to a nearby health facility is a big problem in seeking medical advice or treatment for themselves when they are sick. In our opinion, further operational definition is not required for this particular variable and it is straight forward. 

Reviewer #2 comment: Could you please provide the used appropriate statistical analysis to control and adjust of the effects of confounding factors?

Authors’ response: again many thanks. As we utilized STATA software to run the regression commands, the software will automatically reject if there are confounders because of STATA is different and robust package in auto removing confounding variables. In addition to this, to nullify confounding factors, we utilized bivariable and multivariable decomposition analysis and variable having p-value less than 0.2 are incorporated into the multivariable model and those greater than or equal to 0.2 were not utilized for multivariable. Therefore, utilization of multivariable analysis is one of the mechanism to remove confounding effect.

Discussion

Reviewer #2 comment: Please clarify the methodological limitations of study with scouse on risk of biases.

Authors’ response: many thanks for your comment. More things regarding bias are added to the manuscript (see the new version of this manuscript).

Reviewer #2 comment: Could you please provide application of study findings in education, policy making and future research.

Authors’ response: thank you. Additional explanation were boosted to the revised manuscript according to your comments.

---

## [Decision Letter · Decision Letter 1]

6 Jul 2022

PONE-D-21-17862R1Trends and Predictors of Change of Unmet Need for Family Planning among Reproductive Age Women in Ethiopia , based on Ethiopian Demographic and Health Surveys from 2005-2016: Multivariate Decomposition AnalysisPLOS ONE

Dear Dr. Tareke,

Thank you for submitting your manuscript to PLOS ONE. After careful consideration, we feel that it has merit but does not fully meet PLOS ONE’s publication criteria as it currently stands. Therefore, we invite you to submit a revised version of the manuscript that addresses the points raised during the review process.

The reviewers are now satisfied with your manuscript and their comments are below. However we request you thoroughly copyedit your manuscript for language usage, spelling, and grammar. If you do not know anyone who can help you do this, you may wish to consider employing a professional scientific editing service.  Whilst you may use any professional scientific editing service of your choice, PLOS has partnered with both American Journal Experts (AJE) and Editage to provide discounted services to PLOS authors. Both organizations have experience helping authors meet PLOS guidelines and can provide language editing, translation, manuscript formatting, and figure formatting to ensure your manuscript meets our submission guidelines. To take advantage of our partnership with AJE, visit the AJE website (http://learn.aje.com/plos/)

for a 15% discount off AJE services. To take advantage of our partnership with Editage, visit the Editage

website (www.editage.com) and enter referral code PLOSEDIT for a 15% discount off Editage services. If

the PLOS editorial team finds any language issues in text that either AJE or Editage has edited, the service

provider will re-edit the text for free.

· The name of the colleague or the details of the professional service that edited your manuscript

· A copy of your manuscript showing your changes by either highlighting them or using track changes

(uploaded as a *supporting information* file)

· A clean copy of the edited manuscript (uploaded as the new *manuscript* file)

We look forward to receiving your revised manuscript.

Kind regards,

Thomas Phillips

Staff Editor

PLOS ONE

Journal Requirements:

Reviewers' comments:

Reviewer's Responses to Questions

**Comments to the Author**

1. If the authors have adequately addressed your comments raised in a previous round of review and you feel that this manuscript is now acceptable for publication, you may indicate that here to bypass the “Comments to the Author” section, enter your conflict of interest statement in the “Confidential to Editor” section, and submit your "Accept" recommendation.

Reviewer #1: All comments have been addressed

Reviewer #2: All comments have been addressed

2. Is the manuscript technically sound, and do the data support the conclusions?

Reviewer #1: Yes

Reviewer #2: Yes

3. Has the statistical analysis been performed appropriately and rigorously? 

Reviewer #1: Yes

Reviewer #2: Yes

4. Have the authors made all data underlying the findings in their manuscript fully available?

Reviewer #1: Yes

Reviewer #2: Yes

5. Is the manuscript presented in an intelligible fashion and written in standard English?

Reviewer #1: Yes

Reviewer #2: Yes

6. Review Comments to the Author

Reviewer #1: (No Response)

Reviewer #2: (No Response)

7. PLOS authors have the option to publish the peer review history of their article (what does this mean?). If published, this will include your full peer review and any attached files.

Reviewer #1: No

Reviewer #2: **Yes: **Dr Zeinab Hamzehgardeshi, BMid, MMid, Edu PhD Reproductive Health., Sexual and Reproductive Health Research Center, Mazandaran University of Medical Sciences, Sari, Iran

---

## [Author Response · Author response to Decision Letter 1]

10 Jul 2022

Authors’ response to reviews

Title: Trends and Predictors of Change of Unmet Need for Family Planning among Reproductive Age Women in Ethiopia , based on Ethiopian Demographic and Health Surveys from 2005-2016: multivariable Decomposition Analysis

Authors:

Abiyu Abadi Tareke (abiyu20010@gmail.com)

 Ermias Bekele Enyew (Ermiashi@gmail.com)

Berhanu Fikadie Endeabatu (berhanufikadie@gmail.com)

Abiy Tasew Dubale(abiytasewhi@gmail.com)

Habitu Birhan Eshetu (sisay419@gmail.com)

Sisay Maru Wubante(sisay419@gmail.com)

Version: 1 Date: 10 July, 2022

Author’s response to reviews: 

BMC Health Services Research

Point by point response for editors/reviewers comments

Manuscript title: Trends and Predictors of Change of Unmet Need for Family Planning among Reproductive Age Women in Ethiopia , based on Ethiopian Demographic and Health Surveys from 2005-2016: multivariable Decomposition Analysis

Manuscript ID: ccc25ca3-db68-48c6-87d6-65d9d0770161

Dear editor/reviewer:

Dear all,

We would like to give deep gratitude for your constructive and fruitful comments. You valuable comments enhanced content of the manuscript and we got highly experienced of scientific paper writing. The authors accounted each comments and clarification questions of editors and reviewers in a focused way. Our point-by-point responses to each comment and questions are detailed on the following pages. Further, the details of changes were shown under track changes in the supplementary document attached.

Response to Editors and reviewers’ comments

Editors’ request: The name of the colleague who edited your manuscript? 

Authors’ response: The name of the colleague who helped us in editing and improving the English language of our manuscript was Bayley Adane Takele. He has MPH in biostatistics and 5 years’ experience in scientific paper writing specially in health related manuscript writing.

---

## [Editor Report · Decision Letter 2]

26 Jul 2022

Trends and Predictors of Change of Unmet Need for Family Planning among Reproductive Age Women in Ethiopia, based on Ethiopian Demographic and Health Surveys from 2005-2016:  Multivariable Decomposition Analysis

PONE-D-21-17862R2

Dear Dr. Tareke,

We’re pleased to inform you that your manuscript has been judged scientifically suitable for publication and will be formally accepted for publication once it meets all outstanding technical requirements.

Kind regards,

James Mockridge

Staff Editor

PLOS ONE

---

## [Editor Report · Acceptance letter]

9 Aug 2022

PONE-D-21-17862R2 

Trends and Predictors of Change of Unmet Need for Family Planning among Reproductive Age Women in Ethiopia, based on Ethiopian Demographic and Health Surveys from 2005-2016:  Multivariable Decomposition Analysis 

Dear Dr. Tareke:

I'm pleased to inform you that your manuscript has been deemed suitable for publication in PLOS ONE. Congratulations! Your manuscript is now with our production department. 

Kind regards, 

on behalf of

Dr James Mockridge 

Staff Editor

PLOS ONE